# Effects of Methionine and Glutathione on Acute Ototoxicity Induced by Amikacin and Furosemide in an Animal Model of Hearing Threshold Decrease

**DOI:** 10.3390/biomedicines13061476

**Published:** 2025-06-15

**Authors:** Marek Zadrożniak, Marcin Szymański, Jarogniew J. Luszczki

**Affiliations:** 1Department of Otolaryngology, Head and Neck Surgery, Medical University of Lublin, 20-090 Lublin, Poland; marek.zadrozniak@umlub.pl (M.Z.); marcin.szymanski@umlub.pl (M.S.); 2Department of Occupational Medicine, Medical University of Lublin, 20-090 Lublin, Poland

**Keywords:** methionine, glutathione, furosemide, amikacin, ototoxicity, hearing loss, hearing threshold decrease

## Abstract

**Background/Objectives:** Aminoglycoside antibiotics and loop diuretics are still the main causes of hearing loss in patients, and no specific prevention is available for this drug-induced ototoxicity. The aim of this study was to compare the protective effects of methionine (MET) and glutathione (GLUT) in terms of the reduction in ototoxicity induced by mixtures of amikacin (AMI, an aminoglycoside antibiotic) and furosemide (FUR, a loop diuretic) in a mouse model in which the hearing threshold decreased by 20% and 50%, respectively. **Methods**: To compare the otoprotective effects of MET and GLUT on AMI- and FUR-induced ototoxicity in mice, an isobolographic transformation of interactions was applied. **Results**: MET, but not GLUT, mitigated the AMI- and FUR-induced hearing threshold changes in mice. Additionally, MET exerted an antagonistic interaction with a combination of FUR+AMI, as the hearing threshold decreased by 50%, and an additive interaction, with a tendency toward antagonism in the model of hearing threshold decreased by 20%. In contrast, GLUT exerted only additive interactions when combined with FUR+AMI for both variant hearing thresholds decreased by 20% and 50%, respectively. Only MET could be a potential otoprotective drug in further prevention of hearing loss induced by AMI and FUR. **Conclusions**: MET is superior to GLUT in mitigating AMI- and FUR-induced hearing threshold decreases in mice. MET could be recommended as an otoprotectant in the prevention of hearing loss in patients receiving AMI and FUR.

## 1. Introduction

Drug-induced ototoxicity is a dose-limiting adverse effect of various drugs, mainly aminoglycoside antibiotics, loop diuretics and chemotherapeutics used in patients. Although the severity of ototoxicity varies among patients, it is highly dependent on the drug dose regimen [1,2] and the coadministration of additional medications [3,4,5]. It is widely known that two groups of drugs, i.e., aminoglycoside antibiotics and loop diuretics, are drugs with a particularly high incidence of ototoxic effects in humans [4,6,7]. Accumulating preclinical studies have confirmed that free radical formation and antioxidant inhibition are mainly responsible for changes in the inner ear, causing injury to the outer hair cells and *stria vascularis* [8,9]. Drug-induced ototoxicity can develop either suddenly or progressively depending on the production of reactive oxygen species (ROS) within the cochlea, subsequently inducing inflammation and cellular damage [10,11,12,13,14].

In experimental studies, aminoglycoside antibiotics usually evoke oxidative stress, resulting in permanent hair cell loss [14,15,16,17,18], while loop diuretics usually cause edema of the *stria vascularis* accompanied by a temporary reduction in endo-cochlear potential [18]. Overproduction of ROS disturbs the intracellular redox balance, producing mitochondrial depolarization, which activates caspase-3, leading to the hearing cells damage as a result of apoptosis and ferroptosis [19,20]. The release of cytochrome c and the activation of caspase-9 and caspase-3 are the main factors responsible for aminoglycoside-induced hearing loss [21]. Of note, loop diuretics enhance ototoxic side effects induced by aminoglycoside antibiotics, and such an ototoxic drug combination is expected in patients who are usually receiving aminoglycosides (i.e., amikacin [AMI]) along with loop diuretics (i.e., furosemide [FUR]) and who are receiving continuous ambulatory peritoneal dialysis (CAPD) for peritonitis [22,23]. Therefore, otoprotective drugs are needed to ensure that the human inner ear will not be irreversibly changed during the course of such drug treatment. At present, the widely used strategy to prevent drug-induced ototoxicity is based on antioxidants. Unfortunately, reports documenting the preventive effects of antioxidants in drug-induced hearing loss are limited, and no recommendations exist regarding protective therapy for hearing loss when both aminoglycosides and loop diuretics are administered together [24,25].

Overwhelming evidence indicates that methionine (MET–an α-amino acid) inhibits ROS processes by directly scavenging free radicals to form methionine sulfoxide [26] and by increasing intracellular stores of glutathione (GLUT) as a result of the cysteine supply for endogenous GLUT synthesis [27,28,29,30]. In in vivo animal models of ototoxicity, it was found that MET can serve as an otoprotectant from various ototoxins, including carboplatin [31], aminoglycosides [15,32,33,34], and cisplatin [35]. The molecular mechanisms of the otoprotective action of MET may be linked to its sulfhydryl reversible antioxidant properties [36]. MET is also a direct antioxidant-free radical scavenger, and MET can act as a cysteine sink to fuel the GLUT pathway [8,35,37].

On the other hand, experimental evidence indicates that GLUT plays a crucial role in the detoxification of aminoglycoside antibiotics [38,39,40,41], and GLUT significantly decreases the cochlear damage caused by amikacin and gentamicin in experimental animals [42,43]. The mechanism of GLUT-mediated protection is mediated by the scavenging of certain ROS generated by aminoglycoside antibiotics in cochlear tissue [42,43]. Moreover, ROS-mediated cochlear damage was observed after cisplatin administration as a consequence of GLUT depletion and decreased antioxidant enzyme activities in rats [44].

The aim of this study was to determine the influence of MET and GLUT on the ototoxicity induced by AMI+FUR in a hearing threshold model in mice using auditory brainstem responses (ABRs). The application of MET and GLUT was important because of the prophylaxis that could be used before the administration of the drugs, which can cause ototoxic side effects. Additionally, the direct comparison of the efficacy of MET and GLUT allows us to preselect the most active drug for the treatment of drug-induced hearing loss in this animal model to determine whether further preclinical studies are warranted.

## 2. Materials and Methods

### 2.1. Animals

A total of 144 male Albino Swiss outbred (14–21 days old) mice were randomly assigned to 24 experimental groups comprising 6 animals. All experimental protocols and procedures described herein were approved by the Local Ethics Committee at the University of Life Sciences, Lublin, Poland (License no. 31/2008). All efforts were made to refine procedures, protect animal welfare, minimize animal suffering, and use only the number of animals necessary to produce reliable scientific data according to the 3Rs rule and the ARRIVE guidelines [45]. Both anesthesia and euthanasia methods used in this study were consistent with the commonly accepted norms of veterinary best practice, as presented in the American Veterinary Medical Association (AVMA) Guidelines for the Euthanasia of Animals. Detailed information about laboratory conditions during and after experiments has been described previously [46].

### 2.2. Drugs

Amikacin (AMI—Biodacyna, Polpharma, Ożarów Mazowiecki, Poland), furosemide (FUR—Furosemidum, Polpharma, Ożarów Mazowiecki, Poland), methionine and glutathione (MET and GLUT from Sigma–Aldrich, Poznań, Poland) were administered systemically (i.p.) as follows: MET and GLUT—60 min before AMI and FUR administration, while AMI and FUR were administered i.p. either 15 min (for a decrease in hearing threshold of 20%) or 30 min (for a decrease in hearing threshold of 50%) before hearing measurement. The hearing threshold of the mice was measured in fully anesthetized animals, as illustrated in Figure 1.

### 2.3. Animal Anesthesia

Mice were anesthetized with an intraperitoneal (i.p.) injection with a ketamine (Ketaset^®^, Zoetis Inc., Parsippany, NJ, USA; 100 mg/kg body weight) and xylazine (Rompun^®^, Bayer AG, Leverkusen, Germany; 10 mg/kg body weight) mixture [47,48,49]. After anesthesia induction, mice were individually placed on a homeothermic heating pad system (Harvard Apparatus, Holliston, MA, USA) to maintain a constant body temperature (37 ± 0.5 °C). Additionally, prior to the auditory brainstem response (ABR) recording session, ophthalmic ointment (Recugel^®^, Bausch and Lomb, Vaughan, ON, Canada) was applied to the eyes to prevent corneal drying. Anesthetic depth was assessed every 15 min. for the duration of the experiment by reflexes (corneal touch, pedal flexion), and the core temperature was monitored with a rectal probe.

### 2.4. Auditory Brainstem Responses (ABRs) and Hearing Threshold Detection

A computer-assisted Interacoustics Eclipse EP15 unit (Middelfart, Denmark) measured and collected auditory brainstem responses (ABRs) in mice, as described previously [50,51]. Briefly, alternating clicks generated to the left ear of every mouse allowed for recording the ABRs via subcutaneous electrodes placed near the ipsilateral pinna, vertex and contralateral pinna, with the ground electrode placed along the trunk. Details about the ABRs and their analysis have been described previously [50,51]. Briefly, the sound level of the stimuli decreased from 90 dB sound pressure level (SPL) to 20 dB SPL in 10 dB steps and finally in 5 dB steps to identify the lowest intensity at which an ABR wave V was detectable (Appendix A). At each sound level, up to 1000 responses were averaged and analyzed. If the repeatability of the recorded wave exceeds 95%, the system automatically starts the stimulation with the next programmed volume level. Stimuli were presented at the rate of 39 Hz and recorded for 10 ms duration. Details about the method of applying stimuli and recording ABRs have been described earlier [52]. Hearing threshold was determined by a single observer, who noted the lowest sound level at which a recognizable waveform was seen on a screen from the highest to the lowest sound levels. Waveforms were confirmed as auditory-induced responses by their decreasing amplitude and increasing latency with decreasing sound intensity of the stimulus and repeatability of at least 95%. Hearing thresholds were assessed twice in fully anesthetized mice: before (pretreatment) and after (post-treatment) the i.p. administration of the drugs (MET or GLUT) either alone or in combination with AMI and FUR. The hearing thresholds were determined by a single observer (blinded to the respective treatment), and during the ABR measurements, the anesthetized animals were maintained on a warming pad to maintain a constant rectal temperature. Changes in the hearing threshold (in %) were calculated by comparing the pretreatment with post-treatment hearing thresholds, as described earlier [50,51]. The hearing threshold decrease (in %) for the ototoxic drugs (AMI and FUR, either alone or in combination) was linearly assessed, from which the hearing threshold decreases of 20% and 50% (HTDD_20_ and HTDD_50_) were calculated, as described earlier [50,51].

### 2.5. Animal Euthanasia

After finishing the experiments, the animals were placed in a special uncharged chamber, and euthanasia was performed by means of carbon dioxide (CO_2_), as recommended elsewhere [53].

### 2.6. Isobolographic Transformation

The impact of MET (500 mg/kg, i.p.) and GLUT (500 mg/kg, i.p.) on the hearing threshold was determined in mice receiving AMI and/or FUR with constant doses of MET and GLUT, from which the experimental decreases in the hearing threshold of 20% and 50% (HTDD_20_ and HTDD_50_) were calculated, as presented elsewhere [50]. Subsequently, the impact of MET and GLUT on AMI- and FUR-induced hearing threshold decreases of 20% and 50%, respectively, was determined using the isobolographic transformation of data, as described earlier [50].

### 2.7. Statistical Analysis

The HTDD_20_ and HTDD_50_ values for AMI and FUR were statistically analyzed with one-way ANOVA followed by the post hoc Holm–Sidak multiple comparison tests. The isobolographically transformed additive and experimental values were statistically compared with the Student’s *t*-test with Welch’s correction as presented earlier [54]. Differences were considered statistically significant if *p* < 0.05. All statistical analyses were performed with GrapPad Prism ver. 7.0 for Windows. Additionally, G*Power ver. 3.1 for Windows was used to compute statistical power and effect size.

## 3. Results

### 3.1. Effect of MET and GLUT on the AMI-Induced Hearing Threshold Decrease in Mice

Both MET (500 mg/kg) and GLUT (500 mg/kg), when administered alone (i.p.), had no significant effect on ABR measurement. Since MET and GLUT (when administered separately) did not affect the hearing threshold in experimental animals, it was impossible to calculate the HTDD_20_ and HTDD_50_ for these drugs. In contrast, AMI injected separately (i.p.) dose-dependently diminished the hearing threshold in mice, from which the reduction in the hearing threshold by 20% and 50% (i.e., HTDD_20_ and HTDD_50_) was calculated (Table 1; Figure 2a,c and Figure 3a,c).

Statistical analysis of the data revealed that the HTDD_20_ and HTDD_50_ values of AMI did not differ among the tested groups when MET (500 mg/kg) and GLUT (500 mg/kg) were added (Table 1). However, GLUT (500 mg/kg) decreased the HTDD_20_ and HTDD_50_ values of AMI by 16% and 6%, respectively (Table 1). In contrast, MET (500 mg/kg) increased the HTDD_20_ and HTDD_50_ values of AMI by 16% and 13%, respectively (Table 1). Similarly, the addition of GLUT (500 mg/kg, i.p.) to the mixture of FUR (30 mg/kg) + AMI did not considerably alter the HTDD_20_ and HTDD_50_ values of AMI (Table 1), although the HTDD_20_ and HTDD_50_ values of AMI were reduced by 11% and 3%, respectively. Adding MET (500 mg/kg, i.p.) to the mixture of AMI+FUR (30 mg/kg) did not significantly affect the HTDD_20_ value for AMI despite a 36% increase in the HTDD_20_ value for AMI (Table 1). In contrast, MET (500 mg/kg, i.p.) added to the mixture of AMI+FUR (30 mg/kg) significantly elevated (by 64%) the HTDD_50_ value of AMI (*p* < 0.05; Table 1). G*Power analysis for F test (one-way ANOVA) with “sensitivity” analysis based on a number of analyzed groups (3), total sample size (54), established power of 0.95, and probability error of 0.05 revealed that critical F(2;51) was 3.1799 and effect size (f) was 0.5509, respectively.

### 3.2. Effects of MET and GLUT on the FUR-Induced Hearing Threshold Decrease in Mice

FUR administered separately (i.p.) dose-dependently diminished the hearing threshold in experimental animals, and the reduction in the hearing threshold by 20% and 50% (i.e., HTDD_20_ and HTDD_50_) was calculated (Table 2; Figure 2b,d and Figure 3b,d).

Statistical analysis of the data revealed that the HTDD_20_ and HTDD_50_ values of FUR did not differ among the tested groups when MET (500 mg/kg) and GLUT (500 mg/kg) were added (Table 2). However, GLUT (500 mg/kg) increased (by 6%) the HTDD_20_ and negligibly decreased (by 0.5%) the HTDD_50_ values of FUR (Table 2). In the case of MET (500 mg/kg), the drug increased the HTDD_20_ and HTDD_50_ values of FUR by 12% and 23%, respectively (Table 2). Similarly, the addition of GLUT (500 mg/kg, i.p.) to the mixture of FUR+AMI (500) did not considerably alter the HTDD_20_ and HTDD_50_ values of FUR (Table 1), although the HTDD_20_ and HTDD_50_ values of FUR either decreased by 11% or increased by 13%, respectively (Table 2). Adding MET (500 mg/kg, i.p.) to the mixture of FUR+AMI (500 mg/kg) significantly increased the HTDD_20_ value for FUR by 63% (*p* < 0.01), and the HTDD_50_ value of FUR by 67% (*p* < 0.001; Table 2). G*Power analysis for F test (one-way ANOVA) with “sensitivity” analysis based on a number of analyzed groups (3), total sample size (54), established power of 0.95, and probability error of 0.05 revealed that critical F(2;51) was 3.1799 and effect size (f) was 0.5509, respectively.

### 3.3. Isobolographic Interactions Between AMI, FUR, MET and GLUT in a Mouse Model of Drug-Induced Hearing Threshold Decrease

The isobolographic transformation of the data revealed that the constant dose of AMI (500 mg/kg) added to the increased doses of FUR produced no significant differences between the experimentally derived and theoretically presumed to be additive HTDD_20_ and HTDD_50_ values when MET or GLUT was added (Figure 4a,b).

More specifically, MET (500 mg/kg) increased the doses of FUR that produced HTDD_20_ and HTDD_50_ in the tested mice (Figure 4a,b). However, statistical analysis of data with the Student’s *t*-test and Welch’s correction revealed no significant difference between FUR values at *p* = 0.1050 for a hearing threshold decrease of 20% (Figure 4a) or at *p* = 0.1107 for a hearing threshold decrease of 50% (Figure 4b). Similarly, GLUT (500 mg/kg) increased the doses of FUR that produced HTDD_20_ and HTDD_50_ in the tested mice (Figure 4a,b). However, statistical analysis of data with the Student’s *t*-test and Welch’s correction revealed no significant difference between FUR values at *p* = 0.5799 for a hearing threshold decrease of 20% (Figure 4a) or at *p* = 0.4326 for a hearing threshold decrease of 50% (Figure 4b). Neither MET nor GLUT significantly elevated the dose of FUR in the mouse hearing threshold decrease model. Although the experimentally derived HTDD_20_ and HTDD_50_ values were graphically above the additive values, the lack of statistical significance indicated that the analyzed interactions among the drugs in the mixture were additive in this experimental hearing threshold decrease model in mice (Figure 4a,b). The constant dose of FUR (30 mg/kg) added to the increased doses of AMI exerted additive interactions when GLUT was coadministered because both the HTDD_20_ and HTDD_50_ values did not differ (*p* = 0.5111 for HTDD_20_ and *p* = 0.0544 for HTDD_50_; Figure 4c,d). In the case of MET, an additive interaction (with a tendency toward antagonism) was observed for HTDD_20_ (Figure 4c), and a simultaneous antagonistic interaction was ascribed to HTDD_50_ (Figure 4d). More specifically, MET (500 mg/kg) increased the dose of AMI, decreasing the hearing threshold by 20% and 50% in the tested mice (Figure 4c,d). The Student’s *t*-test with Welch’s correction revealed no significant difference between AMI values at *p* = 0.0544 for a hearing threshold decreased by 20% (Figure 4c), and a significant difference between AMI values at *p =* 0.0164 for a hearing threshold decreased by 50% (Figure 4d). When GLUT (500 mg/kg) was added, the dose of AMI decreased, resulting in hearing threshold decreases of 20% and 50% in the tested mice (Figure 4c,d). In these cases, statistical analysis of the data with Student’s *t*-test with Welch’s correction revealed no significant difference between the AMI values (points A and G) at *p* = 0.5111 for a hearing threshold decrease of 20% (Figure 4c) or at *p* = 0.0804 for a hearing threshold decrease of 50% (Figure 4d).

## 4. Discussion

This comparative study indicated that MET (500 mg/kg) was more effective than GLUT (500 mg/kg) in protecting animals from AMI-induced hearing threshold decreases of 20% and 50%, respectively. It was reported that the HTDD_20_ and HTDD_50_ values for AMI were greater after MET administration than after GLUT administration. In other words, MET strongly protected the animals against hearing threshold decreases, and thus, higher doses of AMI were required to produce the same ototoxic effects, i.e., the hearing threshold decreased by 20% and 50% (HTDD_20_ and HTDD_50_ values) in experimental animals compared with that of GLUT (Table 1 and Table 2). In the case of GLUT, the drug did not protect the mice against a decrease in hearing threshold because the doses of AMI reflecting the HTDD_20_ and HTDD_50_ values did not differ from those for control animals, and moreover, these values were lower than those for AMI administered alone. This finding suggested that GLUT had no effect on hearing thresholds in experimental animals. Although MET protected the animals against a decrease in hearing threshold, it did not fully reverse the effects produced by a combination of FUR(30)+AMI-induced hearing deficits. Both the HTDD_20_ and HTDD_50_ values for AMI+FUR(30)+MET were greater than those for the control (AMI+FUR(30)+VEH-treated) animals, but they did not attain values comparable to those of AMI+VEH. However, the addition of MET to the combination of AMI+FUR(30) significantly alleviated the ototoxic effects according to the HTDD_50_ values. In this case, MET at least in part reversed the hearing deficits mediated by a constant dose of FUR(30) in animals additionally receiving AMI.

On the other hand, MET but not GLUT had a significant impact on FUR-induced hearing threshold decreases in experimental animals. The joint administration of MET combined with two ototoxic drugs affecting hearing in animals (FUR+AMI(500)) completely reversed the effects produced by a constant dose of AMI (500 mg/kg), and the HTDD_20_ and HTDD_50_ values were significantly greater than those for control (FUR+VEH) animals and comparable to the HTDD_20_ and HTDD_50_ values for FUR+MET. In contrast, the joint administration of GLUT with the combination of FUR+AMI(500) did not affect the HTDD_20_ and HTDD_50_ values, suggesting that GLUT had no impact on FUR+AMI(500)-induced hearing deficits in experimental animals.

Notably, under these preclinical conditions, we tested the drug combinations in two experimental variants. The first variant was based on a fixed dose of FUR (30 mg/kg) added to increasing doses of AMI, and the second variant was based on a fixed dose of AMI (500 mg/kg) coadministered with increasing doses of FUR. In both experiments, MET (but not GLUT) mitigated the effects induced by a constant dose of FUR (30 mg/kg) added to AMI-induced hearing deficits and by a constant dose of AMI (500 mg/kg) added to FUR-induced hearing deficits.

Isobolographic transformation revealed that MET and GLUT exerted additive interactions in terms of decreasing hearing thresholds by 20% and 50%, respectively, when a constant dose of AMI (500 mg/kg) was added to FUR-induced hearing deficits (Figure 4a,b). Similarly, GLUT had an additive effect on decreasing hearing thresholds by 20% and 50% when a constant dose of FUR (30 mg/kg) was added to AMI-induced hearing deficits (Figure 4c,d). In contrast, MET exerted an antagonistic effect (*p* < 0.05) when a constant dose of FUR (30 mg/kg) was added to AMI-induced hearing deficits in 50% of the animals (Figure 4d). Simultaneously, MET exhibited an additive interaction with a tendency toward antagonism when a constant dose of FUR (30 mg/kg) was added to AMI-induced hearing deficits by 20% in animals (Figure 4c). Notably, the antagonistic interaction, which is observed among drugs in ototoxicity experiments, is the most desirable interaction (from a clinical point of view) because higher doses of ototoxic drugs are required to produce the same ototoxic effects (i.e., the hearing threshold decreases by 20% and 50%) in this animal model.

In this comparative study, we demonstrated that MET was more effective than GLUT in restoring hearing threshold decreases in experimental mice. Notably, we conducted multiple comparisons among AMI- and FUR-induced hearing threshold decreases in various possible combinations. Although MET and GLUT have quite similar molecular mechanisms of action (i.e., elimination of ROS and mitigation of oxidative stress at the cellular level), it was unexpected that GLUT was less efficacious than MET in this experimental study. This observation may be ascribed to the low activity of GLUT, which is less active than MET in mice. Perhaps rodents are less sensitive to GLUT in vivo than to MET. It is highly likely that MET is able to considerably attenuate the effects of a constant dose of AMI (500 mg/kg) in the presence of increasing doses of FUR. Additionally, MET only partially alleviated the effects produced by a constant dose of FUR (30 mg/kg) in the presence of increasing doses of AMI in experimental animals, especially for a 50% hearing reduction. Considering the abovementioned findings, the antioxidant properties of MET contribute to the alleviation of hearing deficits in this animal model of drug-induced hearing loss.

Another fact should be borne in mind when explaining the difference in activity between MET and GLUT in this experimental model of drug-induced hearing loss. Since MET and GLUT were administered singly after their i.p. administration, the observed difference may result from any difference in the pharmacokinetic distribution of the drugs in animals. Additionally, a difference in penetration of the drugs (MET and GLUT) through the inner ear’s blood–perilymph barrier should be considered as a potential factor influencing the observed difference between MET and GLUT. Moreover, drug–drug interactions between MET or GLUT and AMI, FUR or anesthetic drugs can be a source of different responses to the ABR stimulation. Briefly, some pharmacokinetic changes in the MET and GLUT content in the mice could be responsible for the observed difference in their activity in this in vivo model. Despite identical effects in in vitro experiments and similar mechanisms of action, both MET and GLUT have probably different pharmacokinetic profiles in experimental animals. Unfortunately, no pharmacokinetic parameters of MET and GLUT were measured in this study, and thus, it is not possible to elucidate the exact role of MET and GLUT. Considering the abovementioned possibilities and alternative explanations of the observed difference in hearing threshold in the mice, more advanced pharmacokinetic studies should shed more light and elucidate our observation, explaining the exact role of MET and GLUT in this model of drug-induced hearing loss in mice.

Previously, it was found that two other antioxidants, namely, N-acetylcysteine and vitamin C, alleviated FUR- and AMI-induced hearing threshold decreases by 20% and 50%, respectively, in experimental mice [50,51]. Moreover, nicotinamide riboside, berberine, taxifolin, resveratrol, naringin, ethyl pyruvate, trimetazidine, salvianolic acid B, mitoquinone, (-)butaclamol, piplartine, N-acetylcysteine mitigated aminoglycoside-induced ototoxicity in various experimental models of ototoxicity [55,56,57,58,59,60,61,62,63,64,65,66,67,68,69].

The major limitation of this study was the acute (single) i.p. administration of the drugs (AMI and FUR alone and in combination with MET or GLUT). Neither chronic oral (p.o.) nor chronic systemic (i.v.) administration of the drugs was performed in this study. However, each drug administered chronically usually undergoes specific metabolic transformations in the organism (i.e., distribution into the target tissues, metabolic changes during the first passage through the liver, and elimination from the organism). Notably, during chronic drug administration, each drug reaches its stable state and reaches a balanced state in the target tissue, which can be observed in patients. Unfortunately, chronic administration of ototoxic drugs could irreversibly impair hearing processes in animals, preventing the measurement of any ABR changes in hearing thresholds during the experiments. In this study, the HTDD_20_ and HTDD_50_ values were measured, which reflected any subtle changes in hearing threshold decrease (by 20% and 50%, respectively) in animals after single (acute) administration of the drugs either alone or in combination (AMI, FUR, MET, or GLUT). We are fully aware of the fact that in humans, aminoglycoside-induced ototoxicity side effects occur within days or weeks after systemic application; thus, the conclusions drawn from our experiments should be restricted to the acute experimental conditions.

Another limitation of this study was irreversible hearing loss in individuals receiving aminoglycoside antibiotics because of their destructive effects on outer hair cells [70], in contrast to reversible hearing loss in subjects receiving loop diuretics because of their destructive influence on sodium, chloride, and potassium ions in the cochlea and endolymphatic fluid of the inner ears [70]. Moreover, some interspecies differences between mice and humans, as another limitation, should be borne in mind when extrapolating and transferring the results from this study to clinical conditions. In this study, experiments were conducted with two pretreatment times, i.e., 15 min for HTDD_20_ and 30 min for HTDD_50_. We confirmed that MET exerts its activity within 15–30 min after its i.p. administration, while GLUT perhaps needs more time to mitigate the decrease in hearing threshold in mice. Although the lack of effects of GLUT in this experimental model of hearing deficits in mice can readily explain the observed difference between MET and GLUT, experimental confirmation in further advanced studies is needed. On the other hand, prolonging the pretreatment time for GLUT to more than 30 min would negatively affect fully anesthetized mice, and there would be no guarantee that GLUT could effectively reduce AMI- and FUR-induced hearing deficits in mice if the pretreatment time was prolonged to 120 min. Additionally, in this study, young animals (14–21 days old) were used to avoid and/or minimize the hearing deficits related to the age of the animals, which could occur in the mice [71]. Lack of pharmacokinetic monitoring and estimation of MET and GLUT content in the inner ears of the experimental animals receiving various drugs (AMI, FUR, anesthetics) may also be a main source of limitations in this study.

To overcome these limitations, we attempted to provide information about the impact of MET and GLUT on the ototoxicity induced in mice receiving AMI and FUR, which can be considered a first step in the evaluation of these otoprotectant drugs in further preclinical in vivo studies.

## 5. Conclusions

It was experimentally confirmed that MET is superior to GLUT in mitigating AMI- and FUR-induced hearing threshold decreases in mice. MET could be recommended as an otoprotectant in the prevention of hearing loss in patients receiving AMI and FUR if the results from this study translate to clinical conditions.

## Figures and Tables

**Figure 1 biomedicines-13-01476-f001:**
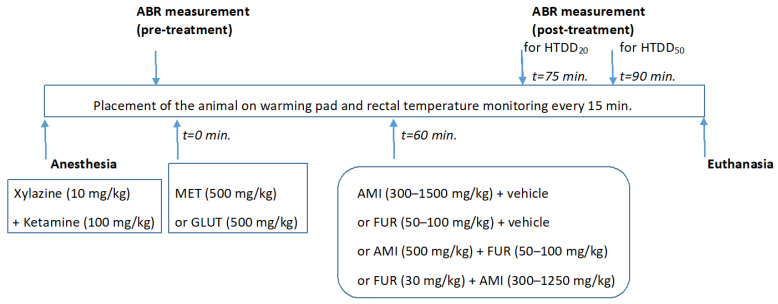
Timeline for induction of anesthesia, *Auditory Brainstem Response* (ABR) measurements, MET or GLUT administration, and AMI and FUR injections in mice. All the drugs used in this study were administered i.p.

**Figure 2 biomedicines-13-01476-f002:**
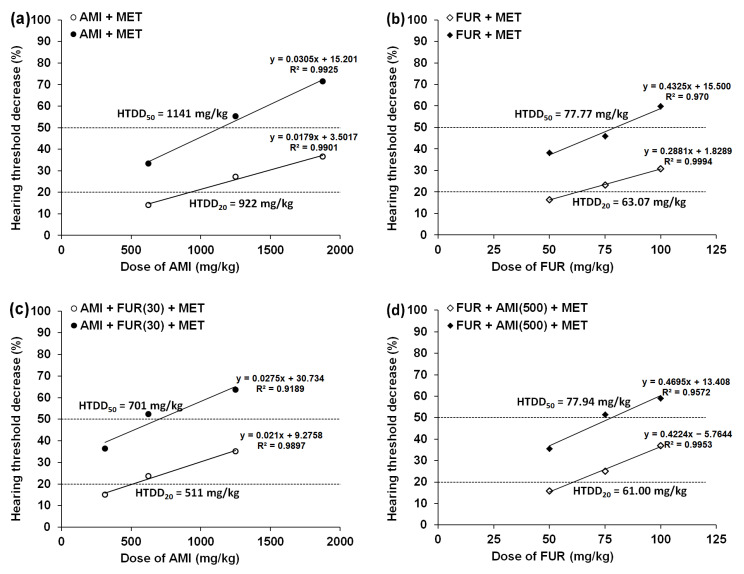
Dose-response relationship lines for MET on AMI- (**a**), FUR- (**b**), (AMI+FUR)- (**c**), and (FUR+AMI)- (**d**) induced hearing threshold decreases by 20% and 50% in experimental animals. Each point on the graphs represents a mean value of 6 animals. To calculate each HTDD_20_ or HTDD_50_ value, 18 mice (3 groups of 6 animals per group) underwent ABR measurements.

**Figure 3 biomedicines-13-01476-f003:**
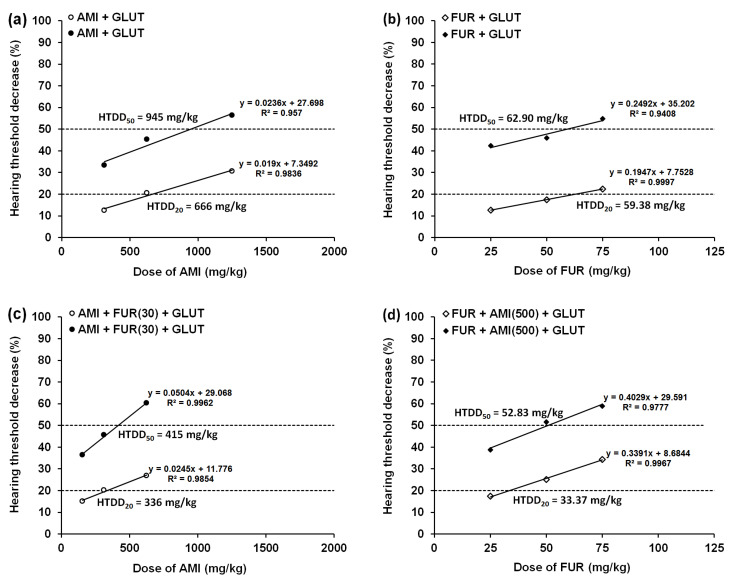
Dose-response relationship lines for GLUT on AMI- (**a**), FUR- (**b**), (AMI+FUR)- (**c**) and (FUR+AMI)- (**d**) induced hearing threshold decreases by 20% and 50% in the experimental animals. Each point on the graphs represents a mean value of 6 animals. To calculate each HTDD_20_ or HTDD_50_ value, 18 mice (3 groups of 6 animals per group) underwent ABR measurements.

**Figure 4 biomedicines-13-01476-f004:**
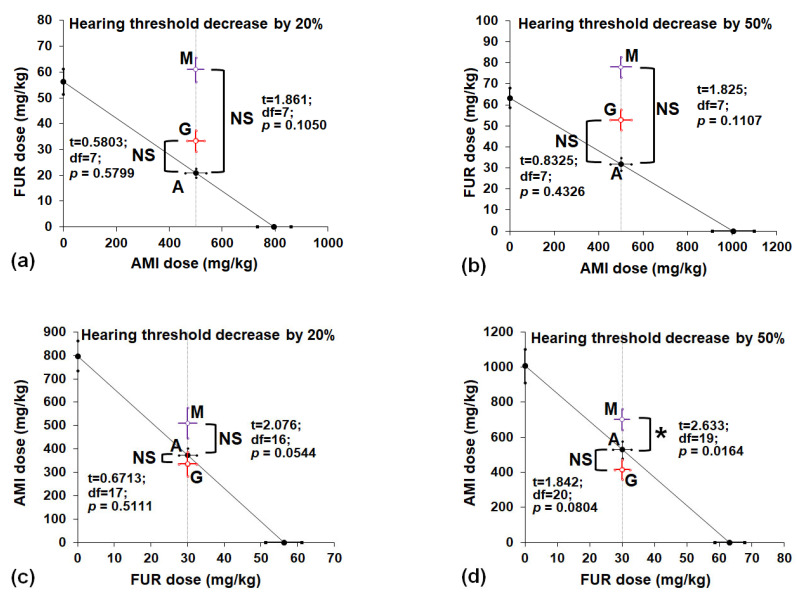
Isobolographic transformation of interactions between AMI, FUR, MET and GLUT in drug-induced hearing threshold decreases by 20% and 50% in mice. The doses of AMI and FUR are plotted on the X- and Y-axes, respectively. The dotted line parallel to the *Y*-axis represents a constant dose of AMI (**a**,**b**) or FUR (**c**,**d**), which was added to increasing doses of FUR or AMI in the mouse model of hearing threshold decreases by 20% and 50%, respectively. Point A (on each graph) illustrates the theoretically additive dose of the two-drug mixture (AMI+FUR) that could produce a hearing threshold decrease of 20% or 50%, respectively. Point M indicates the dose of AMI+FUR+MET that experimentally induced hearing threshold decreases by 20% and 50%, respectively. Point G illustrates the dose of AMI+FUR+GLUT that experimentally induced hearing threshold decreases of 20% and 50%, respectively. * *p* < 0.05 vs. the respective additive value. AMI—amikacin; FUR—furosemide; GLUT—glutathione; MET—methionine; NS—not significant.

**Table 1 biomedicines-13-01476-t001:** Effect of MET and GLUT on the reduction in hearing threshold in animals receiving AMI alone or in combination with a constant dose of FUR.

Treatment	HTDD_20_ (mg/kg)	HTDD_50_ (mg/kg)
AMI+VEH ^a^	797 ± 64	1006 ± 95
AMI+GLUT(500)	666 ± 60 (↓ 16%)	945 ± 88 (↓ 6%)
AMI+MET(500)	922 ± 78 # (↑ 16%)	1141 ± 91 (↑ 13%)
One-way ANOVA	F(2;15) = 3.555	F(2;15) = 1.201
	*p* = 0.0545	*p* = 0.3283
AMI+FUR(30)+VEH ^a^	377 ± 58	428 ± 56
AMI+FUR(30)+GLUT(500)	336 ± 51 (↓ 11%)	415 ± 56 (↓ 3%)
AMI+FUR(30)+MET(500)	511 ± 65 (↑ 36%)	701 ± 60 *,## (↑ 64%)
One-way ANOVA	F(2;15) = 2.466	F(2;15) = 7.926
	*p* = 0.1186	*p* = 0.0045

HTDD_20_ and HTDD_50_ (±SEM) are the doses (in mg/kg) of AMI that reduced the hearing threshold in mice by 20% and 50%, respectively. MET (500 mg/kg, i.p.) and GLUT (500 mg/kg, i.p.) were added to AMI or its combination with FUR (AMI+FUR). Both HTDD_20_ and HTDD_50_ values were statistically analyzed with one-way ANOVA followed by Holm–Sidak’s multiple comparisons test; * *p* < 0.05 vs. AMI+FUR(30)+VEH-treated animals; # *p* < 0.05 vs. AMI+GLUT-treated and ## *p* < 0.01 vs. AMI+FUR(30)+GLUT-treated animals; AMI—amikacin; FUR—furosemide; GLUT—glutathione; MET—methionine; VEH—vehicle; ↓—threshold decrease; ↑—threshold increase. ^a^—results from our earlier study [50].

**Table 2 biomedicines-13-01476-t002:** Effect of MET and GLUT on the reduction in hearing threshold in animals receiving FUR alone or in combination with a constant dose of AMI.

Treatment	HTDD_20_ (mg/kg)	HTDD_50_ (mg/kg)
FUR+VEH ^a^	56.25 ± 4.89	63.24 ± 4.68
FUR+GLUT(500)	59.38 ± 4.92 (↑ 6%)	62.90 ± 4.83 (↓ 0.5%)
FUR+MET(500)	63.07 ± 5.02 (↑ 12%)	77.77 ± 5.01 (↑ 23%)
One-way ANOVA	F(2;15) = 0.477	F(2;15) = 3.074
	*p* = 0.6298	*p* = 0.0761
FUR+AMI(500)+VEH ^a^	37.35 ± 4.10	46.73 ± 4.39
FUR+AMI(500)+GLUT(500)	33.37 ± 4.09 (↓ 11%)	52.83 ± 4.79 (↑ 13%)
FUR+AMI(500)+MET(500)	61.00 ± 4.72 **, ## (↑ 63%)	77.94 ± 4.93 ***,## (↑ 67%)
One-way ANOVA	F(2;15) = 11.99	F(2;15) = 12.34
	*p* = 0.0008	*p* = 0.0007

HTDD_20_ HTDD_20_ and HTDD_50_ (± SEM) are the doses (in mg/kg) of FUR that reduced the hearing threshold in mice by 20% and 50%, respectively. MET (500 mg/kg, i.p.) and GLUT (500 mg/kg, i.p.) were added to FUR alone or in combination with AMI (FUR+AMI). Both HTDD_20_ and HTDD_50_ values were statistically analyzed with one-way ANOVA followed by Holm–Sidak’s multiple comparisons test; ** *p* < 0.01 and *** *p* < 0.001 vs. FUR+AMI(500)+VEH-treated animals; ## *p* < 0.01 vs. FUR+AMI(500)+GLUT-treated animals; AMI—amikacin; FUR—furosemide; GLUT—glutathione; MET—methionine; VEH—vehicle; ↓—threshold decrease; ↑—threshold increase. ^a^—results from our earlier study [50].

## Data Availability

Data are contained within the article.

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
