# Peer review of "Effects of Methionine and Glutathione on Acute Ototoxicity Induced by Amikacin and Furosemide in an Animal Model of Hearing Threshold Decrease"

_biomedicines, 2025, doi:10.3390/biomedicines13061476_

Round 1

Reviewer 1 Report

Comments and Suggestions for Authors

The current research contribution by Zadrożniak et al. is centered on ascertaining the “Effects of methionine and glutathione on acute ototoxicity evoked by amikacin and furosemide in an animal model of hearing threshold decrease”. Overall, the work is timely and relevant considering the intractable nature of chemotherapy-induced hearing loss. It must be mentioned that the study is technically sound, exceptionally well-written and therefore refreshing to read. Authors should be commended for properly articulating a strong rational and clear aims of the research in the Introduction section. There is no doubt that upon publication, this work will constitute a valuable contribution to Biomedicines. Nonetheless, there are a few minor issues which should be addressed in other to improve the quality of the manuscript prior to publication.

Minor comments.

-Please replace ‘evoke’ with either ‘induce’ or mediated’ in all instances across the text.

-Lines 17-19: Please improve the methods used by providing additional details such as how the hearing loss was induced, etc.

-Lines 19-26: Please provide quantitative data to substantiate the effect exerted by MET and GLUT.

-Please adjust Table 1 for clarity.

-The doses of both MET and GLUT (500 mg/kg) used in this study can be considered to be high for the animals. How did authors arrive at this specific dosage and had there been preliminary studies to confirm its potential adverse effect on other vital organ functions and safety in general?

-Line 272: Please change ‘principle’ to ‘principal’ or ‘major”.

-Lines 405-409: Authors should be clear about the replicates used in this study.

Author Response

Minor comments:

Comment:

-Please replace ‘evoke’ with either ‘induce’ or mediated’ in all instances across the text.

Reply:

The term “evoke” has been replaced with “induce” throughout the manuscript

Comment:

-Lines 17-19: Please improve the methods used by providing additional details such as how the hearing loss was induced, etc.

Reply:

Lines: 139-140: Following the second Reviewer’s suggestions, we have inserted a specific citation documenting ABR measurements in rodents ([1] Domarecka, E.; Szczepek, A. J., Universal Recommendations on Planning and Performing the Auditory Brainstem Responses (ABR) with a Focus on Mice and Rats. Audiol Res 2023, 13, (3), 441-458).

Comment:

-Lines 19-26: Please provide quantitative data to substantiate the effect exerted by MET and GLUT.

Reply:

Following the second Reviewer’s suggestions, two figures from the supplementary materials have been placed to the main text of the manuscript illustrating the effects produced by MET and GLUT. MET and GLUT when administered singly had no effects on ABR measurement and thus, had no effect on hearing threshold in experimental animals. Information has been added to the Results section (lines 181-184).

Comment:

-Please adjust Table 1 for clarity.

Reply:

The tables 1 and 2 have been corrected.

Comment:

-The doses of both MET and GLUT (500 mg/kg) used in this study can be considered to be high for the animals. How did authors arrive at this specific dosage and had there been preliminary studies to confirm its potential adverse effect on other vital organ functions and safety in general?

Reply:

The doses of MET and GLUT were experimentally chosen and were high enough to induce the respective changes in the hearing threshold in mice. Since the experiments were performed in an acute model (after single i.p. administration of the drugs), there were no preliminary studies confirming adverse effects of MET and GLUT in mice. We are fully aware of the fact that in a chronic model, MET and GLUT in high doses can induce some morphological changes in mice.

Comment:

-Line 272: Please change ‘principle’ to ‘principal’ or ‘major”.

Reply:

Changed, as suggested

Comment:

-Lines 405-409: Authors should be clear about the replicates used in this study.

Reply:

Following the Third Reviewer’s suggestions we have added a G*Power analysis reporting the size effect, power and probability.

Reviewer 2 Report

Comments and Suggestions for Authors

The manuscript contains interesting data on the prevention of amikacin and furosemide ototoxicity by preliminary administration of drugs with an antioxidant effect. On the one hand, these data allow us to understand the mechanisms of toxicity of amikacin and furosemide, and on the other hand, to approach the development of methods for its drug prevention.

There are small comments that can be considered by the authors as recommendations to improve the article.

  1. It would be appropriate to provide a figure with a schematic representation of the method of applying sound stimuli, methods of recording evoked potentials, etc.
  2. Figure 2 and 3 of the "supplementary materials" section, in my opinion, contains quite valuable information that illustrates the text. It could be placed in the main text of the article.
  3. Oppositely, the text describes in great detail the methods of animal euthanasia. I believe that such a description is redundant. It is sufficient to indicate only the general method of euthanasia and the regulatory documents in accordance with which the experiment was carried out. If the authors consider it necessary to describe euthanasia in such detail, then it would be better to move this section to the supplementary materials.
  4. The authors unexpectedly found that Methionine has a more pronounced preventive effect on ototoxicity than Glutathione. I would like the authors to elaborate on the supposed mechanisms of the effect identified, in particular on possible changes in the pharmacokinetics of methionine and glutathione in living organisms, since these drugs have a similar effect in vitro.
  5. It is also necessary to indicate whether there is antagonism in relation to the main pharmacological effect of the studied drugs and antidotes.

In general, the article is quite novel and is performed at a high methodological level, which allows it to be published after correcting minimal comments.

Author Response

Comment:

It would be appropriate to provide a figure with a schematic representation of the method of applying sound stimuli, methods of recording evoked potentials, etc. Instead of additional figure,

Reply:

In our opinion, there is no need to create a special figures and schematic representations. Instead of this, we have inserted a specific citation reporting in details ABR measurements in rodents ([1] Domarecka, E.; Szczepek, A. J., Universal Recommendations on Planning and Performing the Auditory Brainstem Responses (ABR) with a Focus on Mice and Rats. Audiol Res 2023, 13, (3), 441-458).

Comment:

Figure 2 and 3 of the "supplementary materials" section, in my opinion, contains quite valuable information that illustrates the text. It could be placed in the main text of the article.

Reply:

Both Figures have been placed in the main text of the article as Figures 2 and 3, as suggested.

Comment:

Oppositely, the text describes in great detail the methods of animal euthanasia. I believe that such a description is redundant. It is sufficient to indicate only the general method of euthanasia and the regulatory documents in accordance with which the experiment was carried out. If the authors consider it necessary to describe euthanasia in such detail, then it would be better to move this section to the supplementary materials.

Reply:

Details regarding the animal euthanasia have been deleted, as recommended.

Comment:

The authors unexpectedly found that Methionine has a more pronounced preventive effect on ototoxicity than Glutathione. I would like the authors to elaborate on the supposed mechanisms of the effect identified, in particular on possible changes in the pharmacokinetics of methionine and glutathione in living organisms, since these drugs have a similar effect in vitro.

Reply:

The observed difference between MET and GLUT effects probably results from different pharmacokinetic parameters of the drugs in living organisms (in mice). Since the molecular mechanisms of action of both drugs are quite similar, or even identical in in vitro studies, only differences in pharmacokinetic parameters might be responsible for the observed diverse effects in mice. Unfortunately, we did not measure any pharmacokinetic parameters of MET and GLUT in the experimental mice and thus, we cannot confirm these hypothetical pharmacokinetic changes, as the main explanations for the observed effects. We realize that lack of any pharmacokinetic measurement of MET and GLUT content in mice is the main limitation in this study and we pointed this fact in the Discussion, by adding the respective sentences (lines: 370-388).

Comment:

It is also necessary to indicate whether there is antagonism in relation to the main pharmacological effect of the studied drugs and antidotes.

Reply:

Between furosemide and amikacin there was a strong synergy with respect to the drug-induced hearing loss. Both furosemide and amikacin mutually potentiated the drug-induced hearing loss. There was no antagonism between furosemide and amikacin. In contrast, MET and GLUT partially reversed the furosemide- and amikacin-induced hearing loss. 

Reviewer 3 Report

Comments and Suggestions for Authors

This manuscript presents a well-designed and clearly written study investigating drug-induced ototoxicity and the potential protective effects of metabolic modulators in a murine model. The experimental design is robust, with multiple treatment groups and appropriate controls that allow for a thorough assessment of the interventions. The use of objective measures, such as auditory brainstem response (ABR) thresholds and histological analysis, strengthens the validity of the findings. Moreover, the investigation into metabolic modulators offers relevant and timely insights with potential translational implications for ototoxicity prevention.

The manuscript is generally well-organized, and the logical flow of the experimental rationale and results enhances readability. The figures and tables are informative, although some could benefit from higher resolution and more detailed labeling (e.g., inclusion of error bars and sample sizes).

That said, there are a few areas where the manuscript could be improved:

  1. The statistical analysis section should provide more detailed information regarding the tests used, assumptions checked, and how multiple comparisons were handled.

  2. A justification for the selected sample sizes, ideally including a power analysis, would help support the strength of the conclusions.

  3. While the protective effects of the metabolic modulators are clearly demonstrated, a more in-depth discussion of the potential underlying mechanisms would be valuable.

  4. The current study focuses on short-term outcomes; addressing long-term functional and histological effects would provide a more comprehensive understanding of the treatment’s efficacy.

  5. The discussion could benefit from deeper engagement with recent literature, especially studies exploring similar pharmacological approaches to ototoxicity mitigation.

Overall, this is a valuable contribution to the field, and I believe that, with minor revisions, the manuscript will be suitable for publication.

Author Response

Comment:

The statistical analysis section should provide more detailed information regarding the tests used, assumptions checked, and how multiple comparisons were handled.

Reply:

Lines 176-178: Statistical section has been modified by adding information about statistical software used. The names of statistical tests used in this study have been mentioned in the statistical section.

Comment:

A justification for the selected sample sizes, ideally including a power analysis, would help support the strength of the conclusions.

Reply:

Lines 215-218 and 249-251: We have added G*Power analysis reporting the size effect from F tests and one-way ANOVA. In the G*Power we have selected a “sensitivity” procedure allowing calculating size effect from number of animals used, number of groups analyzed and given probability.

Comment:

While the protective effects of the metabolic modulators are clearly demonstrated, a more in-depth discussion of the potential underlying mechanisms would be valuable.

Reply:

Although molecular mechanisms of MET and GLUT have not been confirmed in experiments in this study, we did not intent to speculate about these mechanisms. Discussion about the potential mechanisms could be only speculative. To avoid misunderstanding and following the second Reviewer’s comments we did not present any speculations and we added an information about such limitations of this study (lines: 430-432).

Comment:

The current study focuses on short-term outcomes; addressing long-term functional and histological effects would provide a more comprehensive understanding of the treatment’s efficacy.

Reply:

We are fully aware of the fact that in a chronic model, MET and GLUT in high doses can induce some morphological changes in mice that could be detected and confirmed histologically. We intent to perform some experiments in a chronic model in a future to confirm the main hypothesis related with ROS, which are responsible for significant changes in hearing threshold after drug-induced hearing loss.

Comment:

The discussion could benefit from deeper engagement with recent literature, especially studies exploring similar pharmacological approaches to ototoxicity mitigation.

Reply:

Line 393: We have added some recent papers reporting ototoxicity mitigation.